# Quantifying First-Order Markov Breakdowns in Noisy Reinforcement Learning: A Causal Discovery Approach

## Abstract

Reinforcement learning (RL) methods often assume that each new observation fully captures the environment's state, ensuring Markovian (one-step) transitions. Real-world deployments, however, frequently violate this assumption due to partial observability or noise in sensors and actuators. This paper introduces a systematic methodology for diagnosing such violations, combining a partial correlation based causal discovery procedure (PCMCI) with a newly proposed Markov Violation score (MVS). The MVS quantifies multi-step dependencies that emerge when noise or incomplete state information disrupts the Markov property.

Classic control tasks (CartPole, Pendulum, Acrobot) are used to assess how targeted noise and dimension omissions affect both RL performance and the measured Markov consistency. Contrary to expectations, heavy observation noise often fails to induce strong multi-lag dependencies in certain tasks (e.g., Acrobot). Dimension-dropping experiments further reveal that omitting certain state variables (e.g., angular velocities in CartPole and Pendulum) substantially degrades returns and elevates MVS, while other dimensions can be removed with negligible effect.

These findings highlight the importance of identifying and safeguarding the most causally critical dimensions to maintain effective one-step learning. By bridging partial correlation tests and RL performance metrics, the proposed approach uniquely pinpoints when and where the Markov property breaks. This framework offers a principled tool for designing robust policies, guiding representation learning, and handling partial observability in real-world RL tasks. All code and experimental logs are publicly available for reproducibility (URL omitted for double-blind review).

## 1 Introduction

Reinforcement learning (RL) typically assumes that observations fully capture the environment's state, ensuring one-step (Markovian) transitions [Sutton and Barto, 1998]. In practice, however, partial observability or sensor noise frequently undermines this assumption [Wisniewski et al., 2024], leading to multi-step dependencies and degraded convergence. While many RL algorithms tolerate mild noise, moderate or poorly structured perturbations often disrupt Markovian structure and erode policy performance.

A key challenge lies in *diagnosing* when (and why) the Markov property ceases to hold. Standard metrics (e.g., final returns) do not reveal whether an environment is effectively "non-Markovian" from the agent's perspective. To address this gap, the present work introduces a *Markov Violation Score* (MVS) derived from partial correlation tests via PCMCI [Runge, 2022]. This score detects significant

lag-2+ dependencies, indicating multi-step effects that deviate from the single-step (first-order) assumption.

A systematic investigation examines how specific perturbations impact both policy performance and MVS in three classic control tasks—*CartPole-v1*, *Pendulum-v1*, and *Acrobot-v1*:

- **Noise Injection.** Gaussian noise and autoregressive noises are applied to observation dimensions at varying levels, revealing which features are critical for stable control.

- **Dimension Dropping.** Entire observation dimensions are removed, forcing learning under incomplete information. Some dropped dimensions cause mild performance degradation, whereas others induce severe instability and high MVS.

- **Markov Violation Analysis.** In each scenario, PCMCI is used to detect higher-lag correlations (lag-2+). Surges in multi-step links typically coincide with sharp performance drops, signaling that first-order Markov assumptions no longer hold.

The results highlight that not all state dimensions contribute equally to preserving Markovian structure. Corrupting or omitting a *critical* variable can produce large multi-step dependencies and abrupt policy collapse, whereas a less influential dimension may have negligible effect. In addition, tasks exhibit distinct thresholds of robustness: some degrade abruptly under moderate noise, whereas others (e.g., Acrobot) handle multi-lag correlations without catastrophic failure.

**Paper Organization.** Sections 2–3.2 discuss related work on partial observability and causal discovery, then introduce the Markov property, PCMCI, and the proposed MVS. Section 5 describes the experimental design (baseline runs, noise injection, dimension dropping) and presents findings on policy performance and Markov consistency. Section 6 addresses limitations and explores directions for future research, and Section 7 concludes the paper.

# 2 Related Works

Real-world reinforcement learning (RL) often encounters partial observability and noisy signals that deviate from the ideal Markov property [wie, 2012]. Much work in *robust RL* aims to handle disturbances in transitions or observations [Panaganti et al., 2022, Liu et al., 2022b], using adversarial training [Pinto et al., 2017] or domain randomization [Wang et al., 2019, Li et al., 2021, Wang et al., 2020] for noisy perception. Other studies introduce noise directly into observations or actions [Hollenstein et al., 2024, Hollenstein et al., Igl et al., 2019], but most evaluations rely on final-return metrics and lack a principled way to detect multi-step dependencies that arise when Markov assumptions fail.

Another branch of *partially observable RL* explores how unobserved variables break Markovian structure [Lauri et al., 2023]. Under POMDPs and related frameworks, latent variables [Liu et al., 2022a] often model environment dynamics [Zhu et al., 2020, Yu et al., Shi et al., 2020]. Though such methods can mitigate certain noise types (e.g., Gaussian or autoregressive (AR) noise ), they seldom track *which* dimensions or episodes are most critical to preserving (or violating) first-order dynamics. Moreover, a single metric to capture multi-lag correlations remains elusive.

Meanwhile, other research [Ota et al., 2020] has *increased* input dimensionality to improve sample efficiency and final performance, reinforcing the need to preserve crucial state information in expanded feature spaces. However, these approaches do not pinpoint *which* dimensions are indispensable for maintaining a Markovian process.

To fill these gaps, the present work applies PCMCI's causal discovery tests [Runge, 2022] to detect higher-lag partial correlations and quantify Markov violations. Building on robust RL's concern with sensor/actuator noise—and partial-observability research on hidden factors—this paper proposes a *Markov Violation Score* (MVS) that aggregates multi-step links beyond first-order transitions. Unlike prior causal-discovery [Zeng et al., 2023] or partial-observation works, the MVS offers a single interpretable value indicating how strongly the Markov property breaks under dimension-dropping or other perturbations. This framework thus moves beyond final-return comparisons to identify *which* omitted dimensions or noise distributions most severely degrade first-order RL learning.

## 3  Preliminaries

### 3.1  Markov Property and Markov Decision Processes

A discrete-time stochastic process $\{X_t\}_{t=0}^{\infty}$ satisfies the *Markov property* if, at every time step $t$, the future state $X_{t+1}$ is conditionally independent of all prior states $\{X_0, X_1, \ldots, X_{t-1}\}$ given the current state $X_t$. Formally,

$$P\big(X_{t+1} \mid X_t, X_{t-1}, \ldots, X_0\big) \;=\; P\big(X_{t+1} \mid X_t\big).$$

Intuitively, this means the present state fully encapsulates all relevant information from the past. In a reinforcement learning (RL) context, we typically apply the Markov property to a state variable $S_t$. If the environment truly satisfies this property, then

$$P\big(S_{t+1} = s', R_{t+1} = r \mid S_t = s, A_t = a, \ldots, S_0, A_0\big) \;=\; P\big(S_{t+1} = s', R_{t+1} = r \mid S_t = s, A_t = a\big),$$

which ensures that only the current state $S_t$ and action $A_t$ determine the distribution over next states $S_{t+1}$ and rewards $R_{t+1}$. However, if noise or partial observability reduce the completeness of $S_t$, higher-order (multi-lag) dependencies may arise. This violates the first-order Markov assumption and can complicate RL methods that rely on single-step dynamics.

#### 3.1.1  Conditional Independence and the PCMCI Framework

Two variables $X$ and $Y$ are said to be *conditionally independent* given a set of variables $Z$ if

$$P(X \mid Y, Z) \;=\; P(X \mid Z).$$

In an ideal Markov process, once the current state $S_t$ is known, the future state $S_{t+1}$ becomes independent of all past states $\{S_0, \ldots, S_{t-1}\}$. However, noise or partial observability can introduce multi-lag dependencies, causing $S_{t+1}$ to depend on earlier states $S_{t-2}, S_{t-3}, \ldots$. To detect such higher-order effects, one can examine *partial correlations*, which measure linear associations between $X$ and $Y$ after conditioning on $Z$. Significant partial correlations at lag $\geq 2$ indicate a breakdown of the first-order Markov property.

Constraint-based causal discovery methods, such as the **PC algorithm** [Spirtes et al., 2001], iteratively test for conditional independence and remove edges in a candidate causal graph. *Momentary Conditional Independence (MCI)* extends this testing to time-series data by conditioning on momentary and past information at each time step. Building on MCI, **PCMCI** [Runge, 2022] combines partial-correlation-based tests with the PC procedure to handle high-dimensional time series. In an RL setting, detecting edges at lag 2 or beyond via PCMCI offers direct evidence that single-step conditioning on $S_t$ alone is insufficient, thus revealing violations of the Markov property.

**Relevance to RL and Markov Violations.**  In RL, $S_{t+1}$ often depends on $(S_t, A_t)$ only. Noise or partial observability can generate dependence on $S_{t-2}, S_{t-3}, \ldots$ beyond $S_{t-1}$. By applying PCMCI to agent trajectories, one can quantify the severity of these multi-lag links. Such diagnosis helps explain policy breakdowns and suggests solutions like state augmentation or sensor fusion [Laskin et al., 2020].

### 3.2  PCMCI and the Markov Property

In a strictly Markovian environment, no significant causal links appear at lags beyond one. When PCMCI detects higher-lag correlations, it indicates missing information in $S_t$. After training, rollouts were collected to apply PCMCI across $S_{t-1}, S_{t-2}, \ldots$ to find significant partial correlations at $k \geq 2$. The *Markov Violation Score* (Section 4) summarizes these multi-lag dependencies. Higher scores typically signal greater departure from first-order dynamics, aligning with observed performance drops.

## 4  Markov Violation Score

As noted in Section 3.2, PCMCI can reveal higher-lag dependencies that indicate violations of the first-order Markov property. This section introduces the *Markov Violation Score* (MVS), which quantifies how severely one-step assumptions are broken.

| Child | Parent | Lag | p-val | Part. Corr |
|:-----:|:------:|:---:|:-----:|:----------:|
| *Variable 0 has 6 link(s):* | | | | |
| 0 | 2 | 0 | 0.00000 | -0.833 |
| 0 | 3 | 0 | 0.00000 | -0.621 |
| 0 | 1 | 0 | 0.00000 | 0.566 |
| 0 | 0 | -1 | 0.00000 | 0.423 |
| 0 | 1 | -1 | 0.00000 | 0.109 |
| 0 | 2 | -1 | 0.00000 | 0.079 |

Table 1: An example of PCMCI results for CartPole showing no significant edges (p-value threshold was 0.05) at lag $\leq -2$, consistent with first-order Markov structure in the unperturbed setting.

**Defining the MVS.** Consider $N$ total variables (e.g., state components), a maximum lag $\tau_{\mathrm{max}}$, and a significance threshold $\alpha_{\mathrm{level}}$. For each variable pair $(i, j)$ and lag $|k| \geq 2$, let $\mathbf{val}_{(i,j,k)}$ be the partial correlation at lag $k$, and let $\mathbf{p}_{(i,j,k)}$ be its p-value. The indicator $\mathbb{I}\big(\mathbf{p}_{(i,j,k)} \leq \alpha_{\mathrm{level}}\big)$ is 1 if the p-value is below $\alpha_{\mathrm{level}}$ and 0 otherwise. The MVS then is

$$\mathrm{MVS} = \frac{\sum_{i=1}^{N} \sum_{j=1}^{N} \sum_{k=2}^{\tau_{\mathrm{max}}} (k-1) \big|\mathbf{val}_{(i,j,k)}\big| \big[-\ln\big(\mathbf{p}_{(i,j,k)}\big)\big] \mathbb{I}\big(\mathbf{p}_{(i,j,k)} \leq \alpha_{\mathrm{level}}\big)}{N^2 \sum_{k=2}^{\tau_{\mathrm{max}}} (k-1)},$$

where $(k-1)$ weights longer lags more heavily. If no lag $|k| \geq 2$ links are detected, then $\mathrm{MVS} = 0$.

| Child Var | Parent Var | Lag | p-value | Partial Corr |
|:---------:|:----------:|:---:|:-------:|:------------:|
| *Variable 0 has 4 link(s):* | | | | |
| 0 | 0 | -1 | 0.00000 | 0.663 |
| 0 | 3 | -3 | 0.00000 | -0.281 |
| 0 | 2 | -3 | 0.00000 | -0.078 |
| 0 | 1 | 0 | 0.03875 | -0.003 |

Table 2: Example PCMCI results ($\alpha$ threshold was 0.05) for a noisy CartPole run with $\mathrm{MVS} > 0$.

A nonzero MVS indicates multi-step dependencies that degrade performance in one-step RL algorithms. Larger scores correlate with more severe Markov violations, whereas $\mathrm{MVS} = 0$ means no multi-lag links survive thresholding and the system remains effectively first-order.

# 5    Experiments and Results

This section explores how noise injection and dimension manipulation impact both the Markovian structure of classic RL environments and final policy performance. The following subsections detail the experimental setup, baseline (no-modification) runs, the effects of i.i.d. and autoregressive (AR) noise, and the consequences of dropping specific dimensions. Each analysis leverages both episode returns and the proposed Markov Violation Score (MVS) to reveal whether multi-lag dependencies emerge under different perturbations.

## 5.1    Experimental Setup

Jobs ran under Python 3.11 on six AWS EC2 `c7i.4xlarge` instances (16 vCPU, 32 GiB RAM, AMI `ami-00c257e12d6828491`); each instance completed an identical slice of the sweep in 12 h, yielding an effective 72 CPU-hours.

Every task used `stable-baselines3` PPO defaults: two 64-unit TANH layers, Adam ($3 \times 10^{-4}$), discount $\gamma = 0.99$, GAE $\lambda = 0.95$, entropy 0, value-loss 0.5, clip 0.2, mini-batch 64, four optimisation epochs, and no hyper-parameter tuning, ensuring identical settings across baseline, noise, and drop variants.

Training horizons followed common benchmarks—50 k steps for *CartPole* and *Acrobot*, 450 k for *Pendulum*. After training, 1–2 k extra transitions per run were gathered to compute PCMCI partial correlations and the Markov-Violation Score (MVS).

## 5.2 Random-Seed Protocol and Significance Estimates

Each EC2 worker chose one baseline seed uniformly from $\{0, \ldots, 1000\}$; the selected value initialised the simulator, network weights, and (where relevant) the noise generator, and was re-used across the baseline, Gaussian, AR, and drop variants for that environment. The six workers therefore produced six statistically independent runs per condition. Learning-curve plots report the mean over $n = 6$ runs, with whiskers showing the 95 % confidence interval $\text{CI}_{95} = 1.96\,\sigma/\sqrt{n}$, where $\sigma$ is the across-seed standard deviation.

For Markov-violation analysis, every trained policy generated three additional roll-outs under fresh rollout seeds; PCMCI statistics from those roll-outs were fused with Fisher's method and then averaged over the six baseline seeds, yielding a single MVS $\pm$ $\text{CI}_{95}$ per setting. All CSV logs produced on the cloud were downloaded and aggregated offline; an appendix script reproduces the merge. Unless noted otherwise, figures and tables follow this protocol.

## 5.3 Baseline Performance

A no-modification "baseline" was trained for each environment to verify that the default tasks exhibit effectively Markovian structure. In all three domains, the baseline (indicated by black curves in subsequent figures) converged quickly and maintained top returns, with PCMCI detecting negligible lag-$\geq 2$ correlations (i.e., MVS $\approx 0$). This outcome confirms that the unaltered state representations of *CartPole-v1*, *Pendulum-v1*, and *Acrobot-v1* largely satisfy first-order Markov assumptions.

## 5.4 Gaussian Noise Injection

To evaluate how i.i.d. Gaussian perturbations affect both policy performance and Markov consistency, each observation dimension $o_t^{(i)}$ is augmented by independent draws $\eta_t^{(i)} \sim \mathcal{N}(\mu, \sigma^2)$. The agent is then trained on $\widetilde{o}_t^{(i)} = o_t^{(i)} + \eta_t^{(i)}$ for the targeted dimension $i$. Figure 1 shows mean learning curves for three noise levels across CartPole, Pendulum, and Acrobot. Small variance ($\sigma^2 = 0.02$) leaves returns near baseline, whereas larger noise ($\sigma^2 = 1.0 - 2.0$) sharply degrades performance when critical angles or velocities are corrupted; Acrobot remains comparatively robust. Aggregated learning curves *with 95 % confidence envelopes*, computed from the full set of seeded runs, are provided in Appendix Fig. 7.

### 5.4.1 State-Space Noise Effects and MVS

Although elevated noise ($\sigma^2 \geq 1.0$) clearly degrades rewards (Figure 3), the Markov property remains fairly intact in i.i.d. Gaussian settings: PCMCI rarely uncovers strong lag-$\geq 2$ correlations unless the variance is extremely high. As illustrated in Figure 2a, changes in MVS remain modest for i.i.d. noise in CartPole, revealing that episodic returns can drop substantially even while MVS hovers near zero. These observations suggest that purely independent noise often fails to violate first-order structure, motivating the introduction of correlated (AR) disturbances to elicit stronger multi-lag dependencies.

## 5.5 Autoregressive Noise Injection

To induce more pronounced deviations from the first-order Markov assumption, *autoregressive* (AR) noise is introduced. Let $\{z_t\}$ be a one-dimensional AR($p$) process,

$$z_{t+1} = \sum_{\ell=0}^{p-1} \rho_\ell\, z_{t-\ell} + \epsilon_t, \quad \epsilon_t \sim \mathcal{N}(0, \sigma^2).$$

This hidden variable $z_{t+1}$ is added to designated *observation* dimensions each step, coupling consecutive states and frequently generating lag-$\geq 2$ dependencies. Experiments varying $p$ and $\rho_0$ confirm that higher AR orders and larger coefficients correlate with elevated MVS and significant performance degradation (Figure 4). Thus, while i.i.d. Gaussian noise alone might not break the single-step property, AR noise reliably induces multi-lag correlations and accentuates the link between MVS and poorer returns.

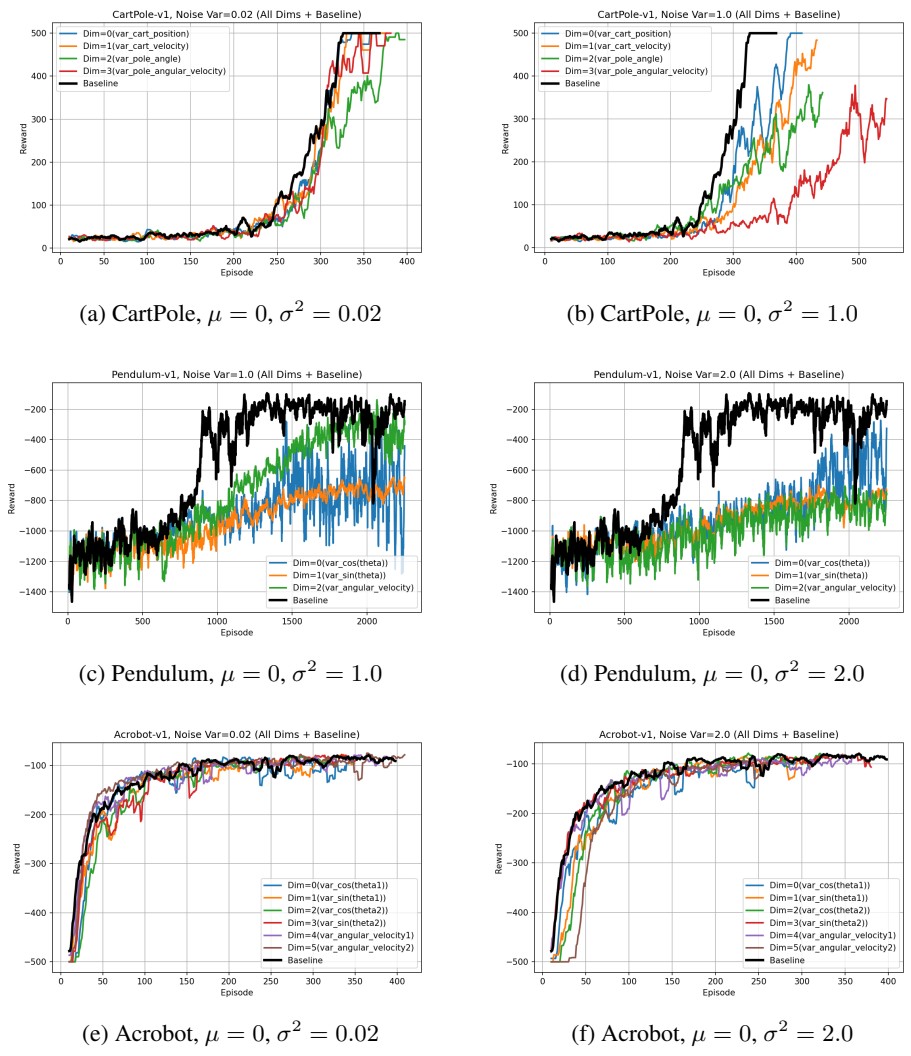

Figure 1: **Effects of i.i.d. Gaussian Noise.** Each panel compares the noise-free *Baseline* (black) to one or more noise-injected settings. **(a,b)** For CartPole, a small variance $(0.02)$ barely disrupts training, but a larger variance $(1.0)$ notably impairs performance when critical dimensions (pole angle or velocity) are perturbed. **(c,d)** Pendulum is more sensitive overall; moderate noise $(1.0)$ already degrades returns, and high noise $(2.0)$ amplifies volatility. **(e,f)** Acrobot remains relatively robust, with minimal slowdowns even at higher noise levels. Overall, certain state dimensions (e.g., angles or angular velocities) are more vulnerable to noise, higher $\sigma^2$ typically delays learning or reduces reward, and the noise-free Baseline continues to provide the fastest and most stable convergence. Aggregated learning curves *with 95 % confidence envelopes*, computed from the full set of seeded runs, are provided in Appendix Fig. 7

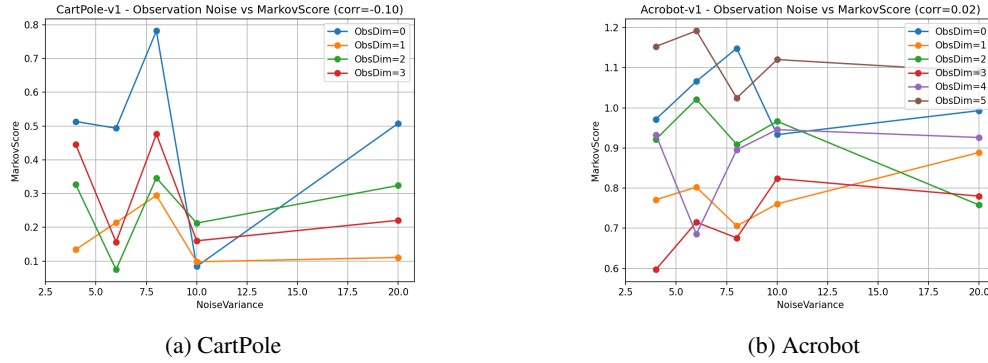

| (a) CartPole | (b) Acrobot |
|:---:|:---:|

Figure 2: **Obs Noise vs. MVS (i.i.d.) in CartPole and Acrobot.** Even with large variance degrading rewards, no strong multi-lag correlations are detected in either environment. For a dimension-by-dimension breakdown across a wider variance grid, see Appendix Fig. 6.

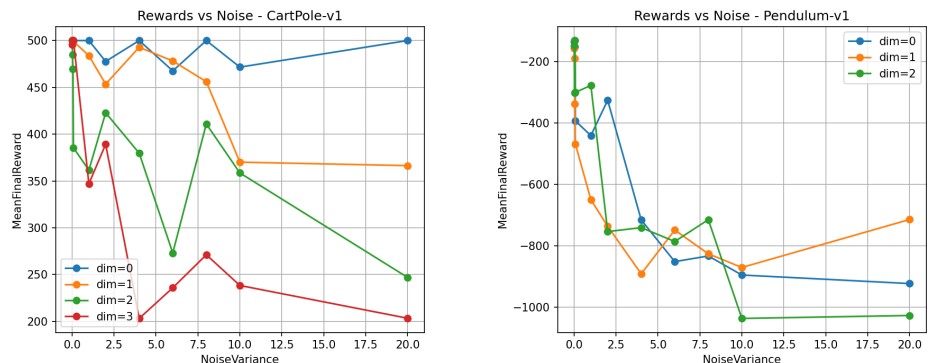

Figure 3: **Rewards vs. noise (i.i.d. Gaussian).** CartPole collapses past moderate noise, while Pendulum degrades more gradually. However, MVS often remains low despite performance drops. For a dimension-by-dimension breakdown across a wider variance grid, see Appendix Fig. 8.

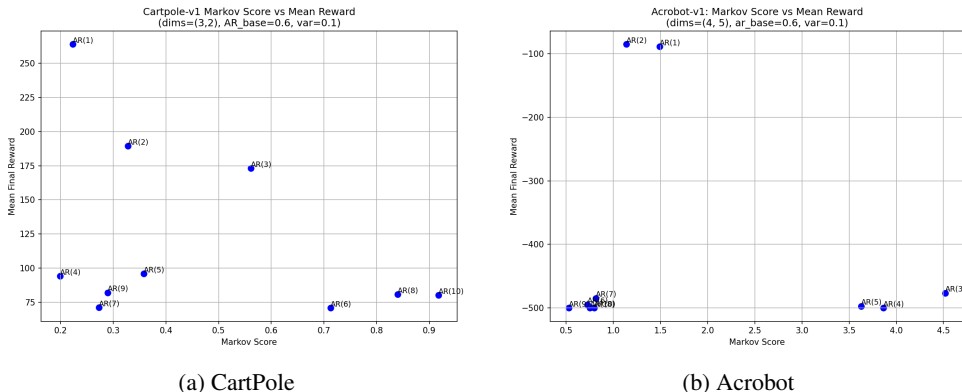

| (a) CartPole | (b) Acrobot |
|:---:|:---:|

Figure 4: **Rewards vs. MVS.** Larger autoregressive orders (ARp) inject correlations, driving up the Markov Violation Score (MVS) and lowering mean final rewards in CartPole and Acrobot. In CartPole, points at lower MVS (e.g. 0.2–0.3) achieve high rewards (200–250), while points at higher MVS (0.7–0.9) sink below 100. In Acrobot, moderate MVS (0.5–1.5) yields near-optimal performance (-100), but values above 3.0 correlate with scores around -400 to -500. A lower-left cluster arises when severe AR noise produces very short episodes, giving PCMCI insufficient data and thus capping the MVS. Overall, stronger ARp disruptions raise MVS and depress returns, revealing how Markov violations undermine one-step RL. Seed-aggregated results with 95 % CIs appear in Appendix Fig. 10.

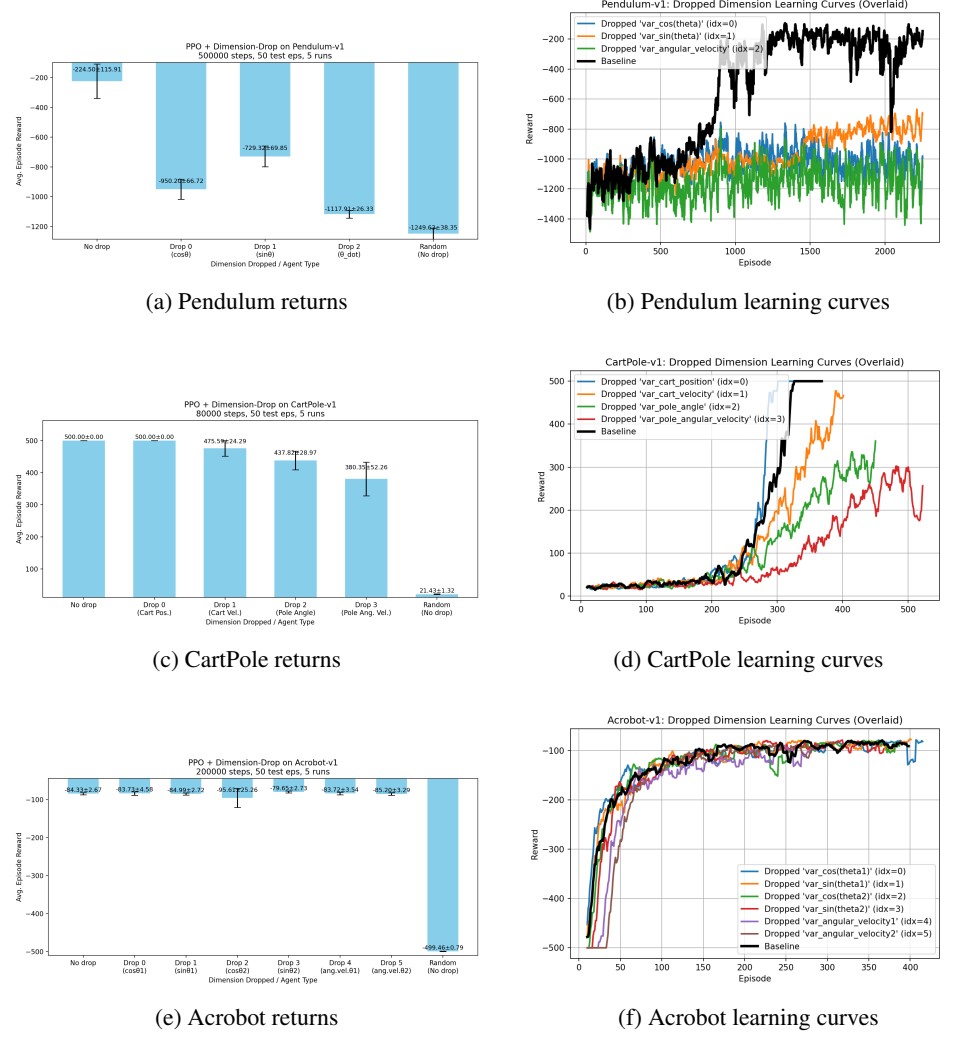

(a) Pendulum returns

(b) Pendulum learning curves

(c) CartPole returns

(d) CartPole learning curves

(e) Acrobot returns

(f) Acrobot learning curves

Figure 5: **Dimension-Dropping Experiments.** Across Pendulum, CartPole, and Acrobot, dropping any single dimension from the observation space degrades performance compared to the Baseline, albeit to varying degrees. The 'Random (no drop)' bar shows returns when no dimension is removed but actions are chosen uniformly at random, serving as a low-baseline reference. Pendulum's average returns (less negative is better) become noticeably worse when dimensions such as $\cos(\theta)$ or angular velocity are omitted, while CartPole's returns (with a maximum of 500) fall sharply if the dropped dimension is pole angle or pole angular velocity. Acrobot, in contrast, remains comparatively robust, showing only minor changes in returns. These differences indicate that each environment depends uniquely on certain state variables for effective control, with Pendulum and CartPole hinging on angular and velocity information. Overall, removing crucial dimensions can significantly impair learning, highlighting the importance of these variables. Sensitivity also varies by environment, as Pendulum and CartPole exhibit steeper performance drops, whereas Acrobot tolerates dimension loss relatively well. Aggregated learning curves across multiple runs with 95 % confidence intervals are shown separately in Fig. 9.

## 5.6 Dimension Dropping

Motivated by the i.i.d. Gaussian noise results (Figure 1), which hinted that certain dimensions were more critical, the next step was to drop each dimension entirely and observe any performance shifts. This is akin to senor malfunction or total loss of signal in real world systems. Surprisingly, removing some dimensions (e.g., cart position in CartPole or various joint components in Acrobot) produced negligible changes, revealing a degree of redundancy in those tasks. In contrast, omitting more pivotal features—such as pole angle or angular velocity in CartPole, or the angular velocity in Pendulum—triggered substantial performance drops and elevated Markov Violation Scores. These findings confirm that although certain state variables can be safely excluded, others are indispensable for first-order RL methods to operate effectively.

## 5.7 Overall Analysis: Correlating MVS and Policy Performance

Collectively, these experiments show how MVS correlates with (and often predicts) policy break-down. In *CartPole* and *Pendulum*, large perturbations to crucial dimensions (e.g., pole angles or angular velocities) often raise MVS and reduce returns drastically; by contrast, *Acrobot* exhibits greater redundancy, tolerating moderate distortions or dropped variables without catastrophic failure. Monitoring MVS alongside standard reward curves thus flags emergent multi-lag dependencies in non-Markov settings (e.g., under AR noise or dimension-critical omissions). Such insights can guide robust controller design and inform representation learning, ensuring that the most causally pivotal features remain intact for stable, first-order RL.

# 6 Limitations and Future Work

These experiments illuminate how noise and dimension manipulations can undermine Markov properties in standard control tasks, yet several constraints remain. Only three benchmarks (*CartPole*, *Pendulum*, *Acrobot*) were examined, limiting applicability to more complex domains. All studies were conducted with PPO—chosen for its empirical stability—which leaves open whether value-based agents (e.g., DQN [Mnih et al., 2013]), entropy-regularised actor–critics such as SAC [Haarnoja et al., 2018], or model-based planners exhibit comparable sensitivities. Noise and dimension perturbations were deliberately simple, and the Markov-Violation Score (MVS) currently relies on linear partial correlations; richer, nonlinear tests could reveal subtler dependencies. Real-world sensor faults and actuator delays also remain unexplored. Future work will target higher-dimensional domains (e.g., multi-joint robotics) that may expose new forms of Markov violation, extend the study to alternative algorithms to test algorithm-level generality, integrate adaptive mitigation (recurrent, Bayesian, or active dimension selection) to suppress MVS spikes, and validate findings in hardware where noise processes are more complex. Planned ablations include injecting irrelevant white-noise channels, evaluating MVS across random/early/final policies, and hard-muting a sensor mid-training to investigate whether MVS can serve as a real-time anomaly detector. *MVS likewise offers a lightweight safety diagnostic for real robots, though treating a low score as a blanket guarantee could prove risky.*

# 7 Conclusion

This work examined how partial observability and noise injection affect Markovian assumptions in reinforcement learning, with a particular focus on detecting multi-lag dependencies through the Markov Violation Score (MVS). Classic control tasks demonstrated that certain dimensions—such as pole angles in CartPole or angular velocity in Pendulum—are pivotal for preserving first-order dynamics, whereas other variables can be removed with minimal impact. Independent Gaussian noise often degraded performance yet did not necessarily induce strong lag-$\geq 2$ correlations, while autoregressive processes consistently triggered higher MVS values and more severe policy break-downs. Dimension-dropping experiments further revealed that some tasks, like Acrobot, retain robustness under omitted variables, whereas others rely heavily on specific state components. These findings highlight the utility of partial-correlation tests for diagnosing Markov violations, indicating the potential for adaptive or model-based methods to mitigate these effects. Extending MVS-based diagnostics to higher-dimensional domains and real-world sensor data offers a promising avenue for developing more robust and generalizable RL algorithms.

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

# A    Implementation & Reproducibility Details

**Hardware.**    Training jobs were distributed over six AWS EC2 `c7i.4xlarge` instances (16 vCPU, 32 GiB RAM, 4th-Gen Xeon; AMI `ami-00c257e12d6828491`). Each instance executed an identical slice of the sweep and finished in $\approx 12$ h, yielding $6 \times 12 = 72$ CPU-hours of compute. Post-processing and figure generation were run locally on an Apple M3 Pro laptop.

**Software environment.**

- Python 3.11.2
- `stable-baselines3` 2.3.0 (policy optimisation)
- `gymnasium` 0.29.1 (environments)
- `tigramite` 5.2.3 (PCMCI causal discovery)
- `numpy` 1.26.4, `scipy` 1.12, `matplotlib` 3.8

A fresh conda (or venv) install can be reproduced with:

```
conda create -n markov python=3.11 -y
conda activate markov
pip install stable-baselines3==2.3.0 gymnasium==0.29.1 \
            tigramite==5.2.3 matplotlib==3.8
```

**Script entry-points.**    All functionality is exposed through a single orchestrator:

```
python markovianess/main.py                    # run the full pipeline
python markovianess/main.py --env CartPole-v1  # one environment only
```

The orchestrator reads a human-readable `config.json` file (specified with `-config_path`) that lists *(i)* environments, *(ii)* training budgets, and *(iii)* noise/perturbation grids. An abridged example is shown below (the full version is included in the supplementary ZIP):

```
{
  "environments": [
    {"name":"CartPole-v1", "time_steps":30000,
     "observations":["CartPos","CartVel","PoleAngle","PoleAngVel"],
     "n_envs":1}
  ],
  "noise_strategies": {
    "gaussian": [
      {"mean":0.0, "variance":0.01},
      {"mean":0.0, "variance":0.05}
    ],
    "auto_regressive": {
      "AR(1)":[{"alphas":[0.9], "sigma":0.1}],
      "AR(2)":[{"alphas":[0.9,0.1], "sigma":0.1}]
    }
  }
}
```

**Training hyper-parameters.**    Across *all* conditions the PPO defaults from `stable-baselines3` were used: two 64-unit TANH layers, Adam learning-rate $3 \times 10^{-4}$, discount $\gamma = 0.99$, GAE $\lambda = 0.95$, clip 0.2, entropy 0, value-loss 0.5, mini-batch 64, and four optimisation epochs per update. **No hyper-parameter tuning** was performed.

**Seed protocol.**    Each EC2 worker drew one baseline seed uniformly from $\{0, \dots, 1000\}$; that seed initialised Gymnasium, network weights, NumPy, Python's `random`, and (where applicable) the noise generator, and was re-used for all perturbations of the same environment on that worker. Accordingly, every condition (baseline, each noise level, each dimension drop) has six statistically independent replicas. Rollout seeds for PCMCI diagnostics were re-sampled independently for every analysis run.

376 **Running time.**   A full sweep over all three environments (*CartPole*, *Pendulum*, *Acrobot*) and all
377 perturbation grids completes in $\approx 12$ h wall-clock, matching the single instance runtime thanks to
378 six-way parallelism.

379 **Top-level workflow (one EC2 worker).**

```
for env in [CartPole, Pendulum, Acrobot]:
    seed = run_baseline(env)                  # clean PPO + PCMCI
    run_noised_gaussian(env, seed)            # i.i.d. obs noise
    run_noised_auto_regressive(env, seed)     # AR(p) obs noise
    run_dropped(env, seed)                     # drop one obs dimension
    collect_results(env)                       # rewards, MVS, plots
```

386 Conceptually, each call above does

387   1. wrap the Gymnasium environment with the requested perturbation,

388   2. train PPO for the budget in `config.json`,

389   3. record the reward curve,

390   4. run $3-5$ extra roll-outs, estimate MVS with PCMCI, and

391   5. save all CSV files and figures under `results/$ENV/`.

392 All perturbations reuse the *same* baseline seed and hyper-parameters, so reward–vs–MVS comparisons
393 are fair and reproducible.

# B   Additional Plots

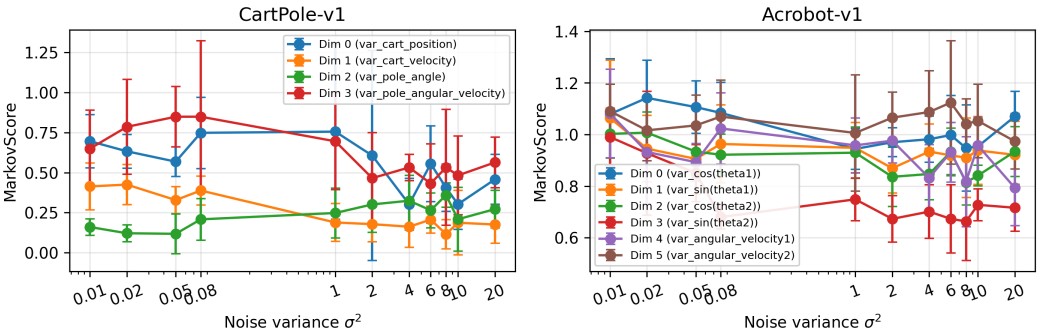

Figure 6: **Markov score under Gaussian noise.** Each colored line tracks the mean Markov-Violation
Score (MVS) over the *six* training seeds (error bars show $\pm 95\,\%$ CI) as i.i.d. Gaussian variance
increases.

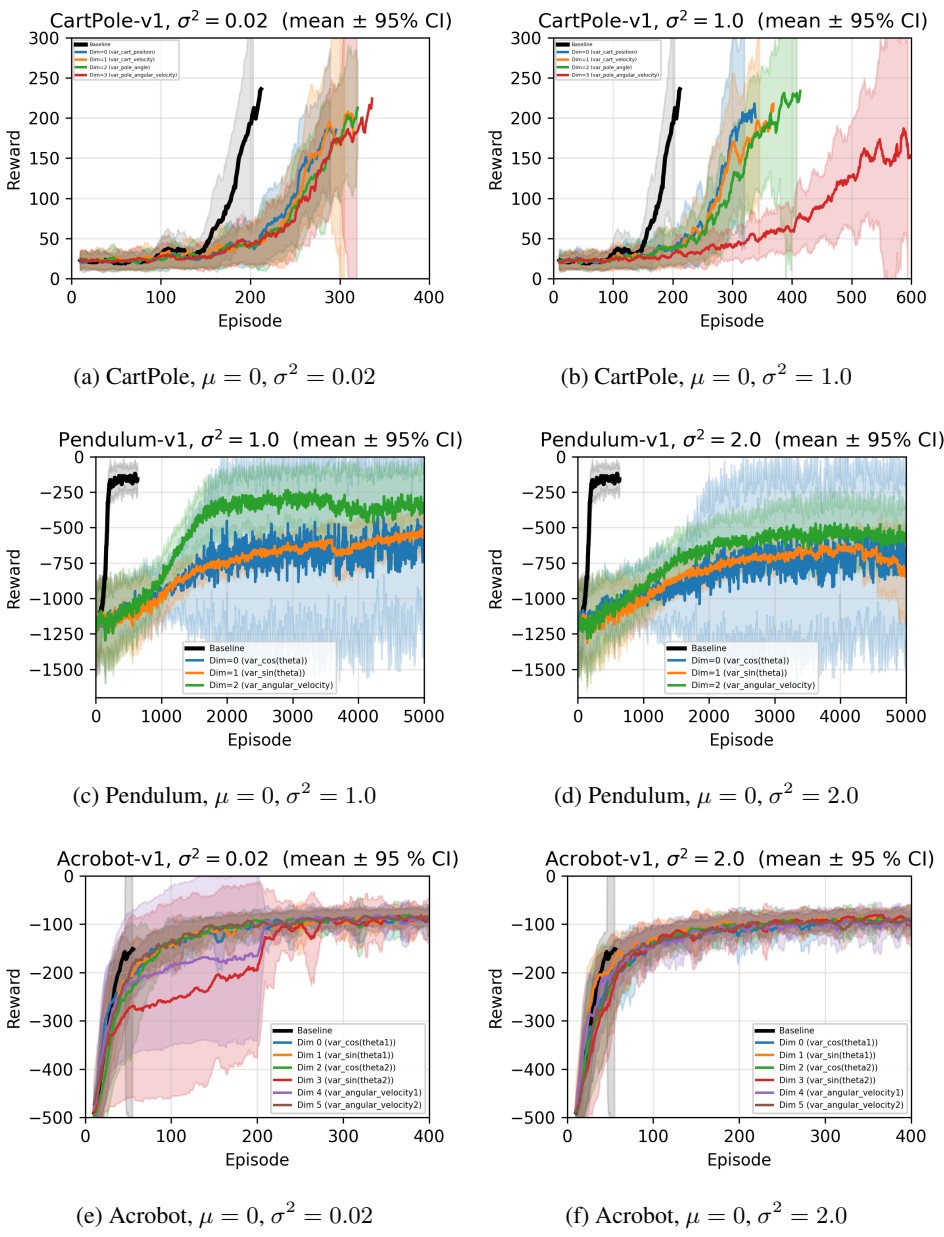

Figure 7: **Gaussian noise with confidence envelopes.** Each panel reproduces the learning curves of Fig.1 but now overlays the *mean ± 95 % confidence interval* (shaded band) obtained from the six independent seeds in §5.2. Solid lines track the seed-averaged episode return; translucent ribbons show sampling variability. For visual clarity, the *Baseline* curve is truncated avoiding a dominant CI band in later episodes. As variance increases—from $\sigma^2 = 0.02$ (light) to $\sigma^2 = 2.0$ (dark)—mean performance drops and uncertainty widens, while *Acrobot* remains comparatively robust. These statistically grounded trends corroborate the qualitative ordering reported in Fig.1.

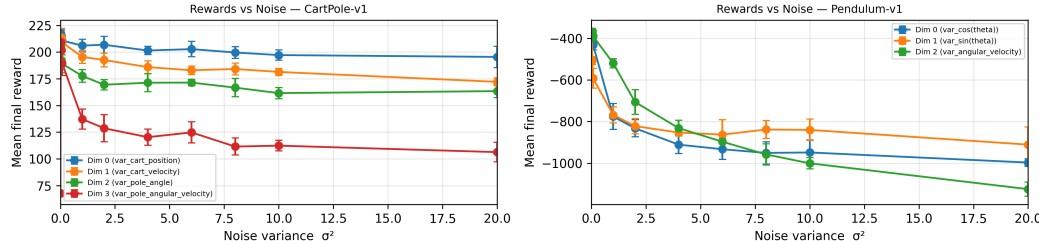

Figure 8: **Mean final reward under Gaussian noise.** Averaged episode return (±95 % CI across six seeds) as a function of noise variance. Left: *CartPole-v1* collapses past moderate noise on dim 3; right: *Pendulum-v1* rewards decline gradually.

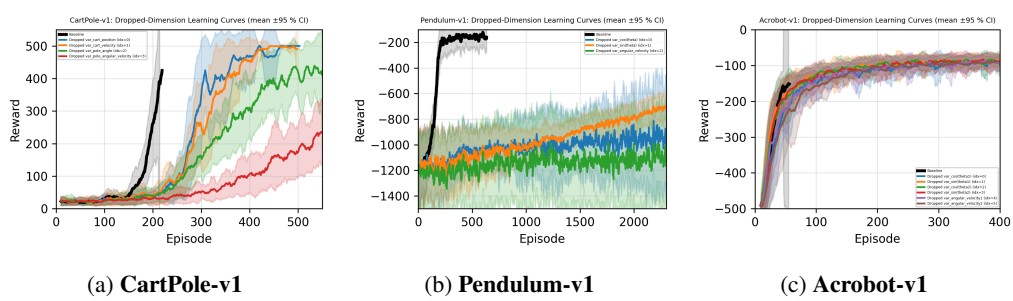

(a) **CartPole-v1**    (b) **Pendulum-v1**    (c) **Acrobot-v1**

Figure 9: Learning-curve comparison when *dropping individual observation dimensions*. Each colored line is the mean reward over episodes after removing the indicated dimension; shaded bands denote 95 % confidence intervals across six runs. For visual clarity, the *Baseline* black curve is truncated avoiding a dominant CI band in later episodes.

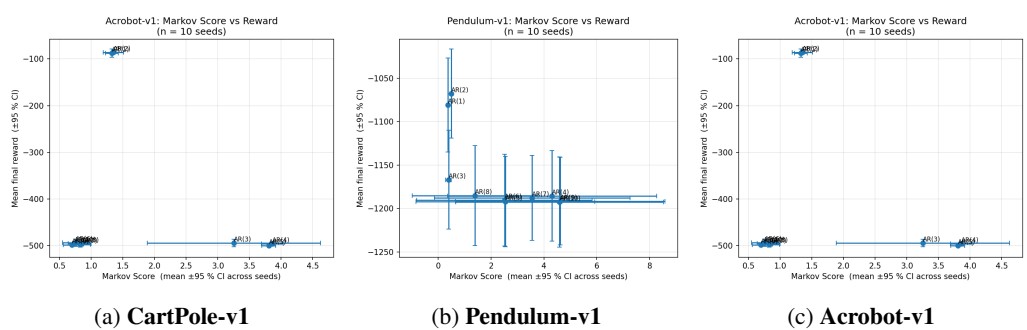

(a) **CartPole-v1**    (b) **Pendulum-v1**    (c) **Acrobot-v1**

Figure 10: **Markov violation versus policy return under autoregressive noise.** Each panel aggregates *ten* independent seeds for every AR-noise setting in its respective environment. Markers report the seed-averaged *Markov-Violation Score* (abscissa) against the corresponding mean final reward (ordinate); error bars denote ±95% CIs across the ten seeds. A clear negative trend emerges: as multi-step dependence intensifies (larger MVS), performance deteriorates. The effect is most pronounced in **CartPole-v1** and **Acrobot-v1**, where high-MVS clusters fall well below their near-optimal low-MVS counterparts. **Pendulum-v1** exhibits the same slope but wider uncertainty because rewards span a broader range and terminate less consistently. Overall, the plot substantiates the key claim that stronger departures from the first-order Markov assumption systematically erode one-step RL returns. All runs share the same AR base coefficient 0.6 and variance 0.1; noise is injected into dims (2, 3) for CartPole, dims (4, 5) for Acrobot, dims (0, 2) for Pendulum.

