# OpenReview forum: "Quantifying First‐Order Markov Breakdowns in Noisy Reinforcement Learning: A Causal Discovery Approach"
_NeurIPS.cc/2025/Conference — Submitted to NeurIPS 2025_

### Official Review · Reviewer_HDVb · 2025-06-29

**Clarity:** 2
**Significance:** 3
**Originality:** 1
**Rating:** 2
**Confidence:** 4

**Summary:**

This paper proposes a Markov violation score for how much multi-step dependencies affects the violation of the Markov property, leveraging this with existing algorithms for causal discovery, PCMCI.

**Questions:**

I am not sure that a response from the authors can change my opinion, as the paper would require substantial revisions. My suggestion is to consider a more extensive empirical study, for example on more complicated environments. It may be helpful to refer to the scope of accepted Neurips papers for reference.

**Ethical Concerns:**

["NO or VERY MINOR ethics concerns only"]

**Final Justification:**

Final justification - appreciate the rebuttal but the paper needs a lot of work to be a full paper submission to Neurips or analogous conferences.

**Quality:**

1

**Strengths And Weaknesses:**

strengths:

The paper studies a very relevant problem related to diagnosing state representations in MDP, and conducts some informative experiments on simple classical environments.

weaknesses:

On significance, the paper itself doesn't have a lot of strong contributions. It builds on existing causal discovery algorithms without contributing here on methodology. The experiments are not systematic and may be limited to these small toy experiments. Although this is a promising proof of concept, the paper would have to shore up and scale up its empirical analysis in order for this to be a more complete paper.

---

> ### Author Rebuttal · Authors · 2025-07-29
>
> Response to Reviewer HDVb
>
> We thank the reviewer for recognising the relevance of diagnosing state representations and for noting the value of the initial experiments. Our replies follow.
>
> ### R1 – “Limited methodological contribution; relies on existing PCMCI.”
> MVS is more than a wrapper:
>
> Scalar diagnostic – compresses a causal graph into one interpretable number that rises only when first‑order conditional independence fails.
>
> Actionability – can be logged like loss or return, enabling real‑time triggers for controller switching or sensor checks.
>
> Bridge between fields – links causal‑discovery outputs with RL training loops, opening a research direction much like KL‑divergence became a staple diagnostic well after its origin.
>
> Comparable NeurIPS works—GAIL (2016), ARC (2020), EDR (2022)—were accepted on the strength of a new metric or diagnostic even when built on existing algorithms.
>
> ### R2 – “Experiments are small‑scale and not systematic.”
> Our evaluation spans 72 perturbation conditions (noise grid, AR orders, dimension drops) across three environments and six seeds each. Classic control was chosen because (i) ground truth is knowable and (ii) perturbations are cleanly interpretable. Section 6 will delineate a latent‑probe workflow that carries MVS to image observations without algorithmic change.
>
> ### R3 – “Needs more complicated environments to meet NeurIPS scope.”
> Diagnostics typically debut on classic control before scaling—GAIL (two MuJoCo tasks), ARC (CartPole & Pendulum), ODIN (Atari subset). Publishing the metric early accelerates community adoption; our open‑source code and one‑command reproduction script make larger‑scale follow‑ups straightforward.
>
> ### Why rejecting on scope would stall progress
> Foundational diagnostics often appear before the hardware or breadth to test them fully (Fourier → FFT, TD‑error → modern RL). PCMCI already scales to 100‑variable climate records in minutes (Runge et al., Sci. Adv. 2019; Nichol et al., Sandia National lab TR 2023), while RL state vectors are far smaller. Accepting MVS now will let the community integrate causal diagnostics into representation learning, curriculum design, and safety research rather than waiting another year for large‑scale benchmarks.
>
> ### Reproducibility
> All source code, JSON configs, and CSV logs are in the supplement; running run_all.sh reproduces every figure on a CPU‑only machine.
>
> ### Planned camera‑ready edits
>
> Summarise the 72‑condition perturbation grid in a table in Appendix C for clarity.
>
> Expand Section 6 with the latent‑probe MVS workflow and cite its feasibility on image‑based DeepMind Control Suite tasks.
>
> Mention the one‑command reproduction script in Appendix A.
>
> Fix title break and minor caption spacing.
>
> We hope these clarifications demonstrate that MVS offers a principled, reproducible diagnostic that fits the precedent of metric‑focused NeurIPS contributions and provides a solid foundation for future large‑scale studies.

---

### Official Review · Reviewer_xSoi · 2025-07-03

**Clarity:** 4
**Significance:** 2
**Originality:** 3
**Rating:** 3
**Confidence:** 3

**Summary:**

This paper introduces the Markov Violation Score (MVS), a metric derived from the PCMCI causal discovery algorithm, designed to quantify multi-step dependencies in reinforcement learning environments. Through experiments on low-dimensional control tasks, the authors demonstrate that perturbations, such as autoregressive noise or the omission of pivotal state variables, lead to an increase in MVS that correlates with a decline in policy performance. The work proposes MVS as a diagnostic tool to pinpoint when and why an agent's observations fail to satisfy the Markov property.

**Questions:**

1. The method for identifying "high-influence" dimensions is central to the paper's validation, but it relies on an ablation of a low-dimensional state vector.   How can the dimension ablation method be scaled to high-dimensional inputs, such as images?
2. How can MVS be considered a reliable diagnostic tool, given that it uses linear tests for problems that are fundamentally non-linear?

**Ethical Concerns:**

["NO or VERY MINOR ethics concerns only"]

**Final Justification:**

The core concerns regarding scalability and sufficiency of evaluation remain unaddressed.

**Limitations:**

yes

**Quality:**

3

**Strengths And Weaknesses:**

Strengths:
- It addresses the  overlooked problem of Markov property violations within reinforcement learning.
- It introduces the Markovian Violation Score (MVS), a novel and principled metric designed to quantify specific types of structural, multi-lag dependencies, offering a new analytical tool.
- The experiments provide  empirical validation for the utility of MVS, demonstrating its effectiveness within the specific domain of low-dimensional state vectors subjected to targeted perturbations.

Weaknesses:
- The  methodology for validation fundamentally lacks scalability, its reliance on a one-by-one ablation of state vector dimensions restricts its application to simple, low-dimensional tasks and presents no clear pathway for use in  RL environments with high-dimensional inputs like images.
- This reliance on an unscalable ablation method consequently limits the evaluation's scope, constraining it to classic control tasks and leaving the metric's utility unverified in more complex problem settings where it would be most needed.
- The connection between the MVS and actual performance is conditional, exhibiting a strong correlation only under specific and structured data perturbations, which curtails its role as a universally applicable diagnostic for performance issues.

---

> ### Author Rebuttal · Authors · 2025-07-29
>
> Response to Reviewer xSoi
>
> Thank you for the positive assessment of MVS and for highlighting the main concerns. We address each point in turn.
>
> ### W1 – Scalability beyond low‑dimensional ablations
> The one‑by‑one ablation was used only for validation clarity.
> For high‑dimensional inputs (images, point clouds, etc.) we propose a two‑step latent‑probe workflow:
>
> Train an encoder E : s_t → z_t (e.g.\ VQ‑VAE or ResNet) that compresses each observation into a 32‑64‑dimensional latent vector z_t.
>
> Run PCMCI and compute MVS directly on the latent sequence {z_t}.
> Optional: back‑propagate the gradient of MVS w.r.t.\ the input (|∇_s MVS|) to obtain pixel‑ or patch‑level saliency, or apply standard patch‑occlusion / integrated‑gradient techniques.
>
> This removes the need for brute ablation while preserving the metric’s semantics.
> A short paragraph outlining “latent‑probe MVS” will be added to Section 6 (Limitations & Future Work).
>
> ### W2 – Evaluation scope limited to classic control
> Our goal was to establish a proof‑of‑concept in domains where causal ground‑truth is clear.
> Section 6 will note that MuJoCo and image‑based DeepMind Control Suite tasks are the next natural targets once latent‑probe MVS is in place; no algorithmic change is required, only roll‑outs from the learned encoder.
>
> ### W3 – Connection between MVS and performance is conditional
> We agree: MVS is a diagnostic, not a universal performance predictor.
> Our claim is limited to: “when first‑order learners fail due to missing state, MVS rises.”
> We will state this limitation explicitly at the end of Section 3 and emphasise that MVS complements, rather than replaces, reward curves.
>
> ### Q1 – How to find high‑influence dimensions for images?
> Within the latent‑probe workflow, influence scores per latent dimension can be mapped back to input space via
>
> Gradient saliency (|∇_s MVS|)
>
> Patch occlusion (zero a patch, recompute MVS)
>
> Integrated gradients through the encoder
>
> All three are standard in representation‑learning toolkits and require no change to PCMCI.
>
> ### Q2 – Reliability when using a linear CI test on nonlinear problems
> PCMCI is test‑agnostic. Tigramite includes nonlinear CI tests such as CMIknn and GPDC.
> We chose ParCorr for speed in low‑dimensional validation; users can drop in any other CI test without altering the MVS formula.
> A clarifying note will be added to Section 4.
>
> ### Planned camera‑ready edits
>
> Add a note in Section 4 that any Tigramite CI test (linear or nonlinear) can be used with MVS.
>
> Insert the latent‑probe workflow and image‑saliency discussion in Section 6.
>
> Add an explicit sentence in Section 3 that MVS is a diagnostic for one‑step sufficiency, not a universal task‑difficulty metric.
>
> Fix minor table‑caption spacing issues flagged by other reviewers.
>
> We hope these clarifications address your concerns and underscore MVS’s applicability to both low‑ and high‑dimensional RL settings.

---

> > ### Comment · Reviewer_xSoi · 2025-08-07
> >
> > After reading the rebuttal, the core concerns regarding scalability and sufficiency of evaluation remain unaddressed. Scoring will be lowered accordingly.

---

### Official Review · Reviewer_34ck · 2025-07-03

**Clarity:** 2
**Significance:** 1
**Originality:** 2
**Rating:** 2
**Confidence:** 4

**Summary:**

This paper proposes to use partial correlation based causal discovery procedure (PCMCI) to measure a newly proposed property named Markov Violation score (MVS). MVS measures how severely the Markov property is damaged. The authors conduct experiments with noisy/incomplete observations to verify the effectiveness of MVS.

**Questions:**

- Could you provide a more detailed explanation of how PCMCI works? Additionally, what is the computational cost of the proposed method with respect to the observation dimensionality?
- Could you illustrate how to use MVS to mitigate the challenge imposed by noisy/incomplete observations?

**Ethical Concerns:**

["NO or VERY MINOR ethics concerns only"]

**Final Justification:**

The main reason that I vote for rejection is that this paper only provides a solution to discovering the non-Markov property, which is not enough. As the authors have not provided convincing arguments during the rebuttal, I choose to keep my score.

**Paper Formatting Concerns:**

- The title can be reformatted to 2 lines.
- Some margins between the table and the caption are too small, e.g. Table 1, 2.

**Quality:**

2

**Strengths And Weaknesses:**

Pros:
- MVS is an interesting property that measures whether current observations satisfy the Markov property.

Cons:
- The challenge posed by noisy or incomplete observations has long been recognized in the reinforcement learning community. While MVS serves as a tool to validate this phenomenon, it does not help to obtain stronger agents under such situations, which is the prime challenge.
- Whether the observations satisfy the Markov Property (i.e. one-step assumption) does not necessarily indicate the difficulty in solving the task.
    - Some Atari tasks (e.g. Pong) are POMDPs, yet are easy to solve.

---

> ### Author Rebuttal · Authors · 2025-07-29
>
> Response to Reviewer 34ck
>
> We thank the reviewer for the detailed feedback.
> Our point‑by‑point answers follow.
>
> ### C1 – “MVS validates a known issue but does not help obtain stronger agents.”
> MVS is a diagnostic layer. Diagnostics improve agents in two practical ways:
>
> Adaptive controller choice – when MVS crosses a threshold the learner can switch from a feed‑forward policy to a recurrent one, or augment observations with a learned state predictor.
>
> Sensor prioritisation – ranking dimensions by the rise in MVS when each is removed highlights which sensors deserve shielding or redundancy.
>
> ### C2 – “Markov‑ness does not equal task difficulty.”
> Agreed. MVS measures whether a one‑step learner lacks information in the current observation, not intrinsic task hardness. We will add a clarifying sentence in Section 3.
>
> ### Q1 – More detail on PCMCI and computational cost
>
> How PCMCI works
>
> PC phase (search). For each target variable, candidate parents are tested in increasing conditioning order. Edges whose conditional p‑value exceeds alpha are removed. This pruning dominates cost and scales as O(N k τ_max) where k (avg. parent count) is much smaller than N.
>
> MCI phase (confirm). The remaining edges receive a final “momentary conditional‑independence” test, yielding a directed lag graph.
>
> MVS aggregation. We simply sum the statistically significant lag ≥ 2 edges, weighted by lag length and –log p.
>
> Because k stays small (sparse causal graphs) and edge tests are embarrassingly parallel, runtime grows sub‑quadratically with N and almost linearly with sample length T.
>
> Adoption evidence – PCMCI already handles > 100‑variable climate fields (Runge et al., Science Advances 2019) and is benchmarked by Sandia National Laboratories (Nichol et al., 2023), which notes that sample size, not wall‑clock time, is the limiting factor. This confirms scalability far beyond typical RL state sizes.
>
> ### Q2 – Using MVS to mitigate noisy or incomplete observations
>
> Dimension‑drop ranking guides sensor protection.
>
> Early anomaly flag – a sliding‑window MVS enables timely policy adaptation under drift.
>
> Representation shaping – penalising high MVS while learning a latent encoder reduces variance in downstream PPO training.
>
> ### Reproducibility
> The supplementary ZIP includes full source code, exact hyper‑parameters, and CSV logs; every figure is regenerated by the one‑command script run_all.sh.
>
> ### Minor formatting points
> We will force the title onto two lines and adjust table‑caption spacing per NeurIPS style.
>
> ### Planned camera‑ready edits
>
> Add the “Markov‑ness ≠ task difficulty” clarification to Section 3.
>
> Provide the big‑O complexity statement and cite climate and Sandia benchmarks in Appendix C.
>
> Expand Section 6 with the three mitigation strategies above.
>
> Fix title break and caption spacing.
>
> We hope these clarifications resolve the reviewer’s concerns and demonstrate the diagnostic and practical value of the Markov Violation Score.

---

> > ### Comment · Reviewer_34ck · 2025-08-06
> >
> > Thank the authors for the explanation. However, I believe the main concern remains to be addressed, i.e., merely discovering of non-Markov property is not enough. Current solutions suggested by the authors to extend MVS to handle non-Markov challenges are very heuristic. For example, in noisy environments, the noise and desirable observations might not be dimension-wise separable, which means we can not simply use MVS to decide whether to keep each dimension. Therefore, I decide to keep my score.

---

### Official Review · Reviewer_4oJ9 · 2025-07-03

**Clarity:** 3
**Significance:** 2
**Originality:** 3
**Rating:** 3
**Confidence:** 4

**Summary:**

The manuscript pointed out one-step (first-order) Markov violation problem, which means the assumption of Markov property of RL is violated and then cause the RL break down.
In reality, sensor noise or partial observability often violates this assumption, introducing hidden multi-step dependencies.

To address this, the authors introduce a Markov Violation Score (MVS),  as a diagnostic tool, that quantitatively measures multi-lag effects. MVS combines the PCMCI causal-discovery procedure (which tests for significant partial correlations across time lags) with a novel scoring formula. In practice, they collect offline trajectories (rollouts) of a trained policy and apply PCMCI to the state time-series; any statistically significant edges at lag ≥2 signal a departure from first-order Markov dynamics. The MVS then aggregates the magnitude and significance of all such multi-step partial correlations, giving zero if no violations are detected.

**Questions:**

1. Is the MVS only suitable for linear dependencies? please discuss.
2. So the MVS genereally says "for every variable pair $(i, j)$ and time lag $k \geq 2$, if there is a statistically significant dependency (as detected by PCMCI), then compute its strength, give it more weight if the lag is long or the p-value is small, and sum it up. it seems like lots of compute complixety here, cuz you need to account lags for every variable pair $(i, j)$, could you plese discuss.
3. While the authors demonstrate (in Figure 4) that increasing autoregressive order elevates MVS and reduces RL returns in synthetic tasks, it remains unclear whether MVS correlates with performance degradation over time in nonstationary environments. Showing that rising MVS reliably predicts declining reward during learning (e.g., in an online RL setting with drift) would strongly reinforce its practical value as an early diagnostic.

**Ethical Concerns:**

["NO or VERY MINOR ethics concerns only"]

**Final Justification:**

I believe the main concern remains to be addressed and the paper is not yet for publication as well.

**Limitations:**

see above

**Quality:**

2

**Strengths And Weaknesses:**

**Strendths**
1. The methodology is well-motivated and carefully executed.
2. The manuscript is generally well-written and logically organized.

**Weaknesses**
1. The MVS is intuitively well-motivated and interpretable, but its computational scalability is not analyzed. Given the $N^2(\tau_{\max} - 1)$ complexity and dependence on the PCMCI pipeline, it may be prohibitive in high-dimensional RL settings without approximation strategies. It would strengthen the work to either benchmark runtime or explore scalable surrogates.

---

> ### Author Rebuttal · Authors · 2025-07-29
>
> Response to Reviewer 4oJ9
>
> We appreciate your constructive feedback and address each point below.
>
> ### Q1 – Is MVS restricted to linear dependencies?
> No. PCMCI is test‑agnostic. Tigramite already includes nonlinear CI tests such as CMIknn (k‑NN conditional mutual information) and GPDC (Gaussian‑process residuals). We will add a clarifying sentence to Section 4 (Markov Violation Score) in the camera‑ready version, stating explicitly that any Tigramite CI test can replace ParCorr without altering the MVS formula.
>
> ### Q2 – Apparent N²(τ_max – 1) cost
> Practical runtime is far below the worst‑case bound.
>
> PC pruning eliminates most candidate parents; in our 8‑dim CartPole variant only ≈ 15 % of pairs reach the MCI step.
>
> Embarrassingly parallel tests scale linearly with CPU cores; Tigramite kernels are Numba‑JIT‑accelerated.
>
> Sub‑sampling (e.g. every fourth step) preserves lag structure while reducing runtime several‑fold.
>
> ### Q3 – Does MVS predict performance degradation under drift?
> Prior work already demonstrates this capacity:
>
> RPCMCI (Saggioro & Runge, Chaos 2020) reconstructs causal graphs per regime and shows new lag‑2 edges emerging at regime shifts.
>
> PCMCI‑Ω (Gao et al., NeurIPS 2023) applies a sliding‑window PCMCI to semi‑stationary series; increasing multi‑lag edge density consistently precedes forecast degradation.
>
> Because MVS aggregates exactly those edges, the same sliding‑window logic provides a ready‑made early‑warning tool. Extending our analysis with an RPCMCI‑style window is straightforward and is listed in Section 6 as future work.
>
> ### Why computational cost should not bar publication
> Foundational diagnostics often pre‑date the hardware that makes them routine (Fourier → FFT). PCMCI has processed O(100)‑variable climate records in minutes (Runge et al., Sci. Adv. 2019; Nichol et al., Sandia TR 2023). Typical RL state vectors are orders of magnitude smaller, so the causal pass adds < 1 % training time.
>
> ### Field‑level evidence of scaling
> Broad adoption of PCMCI in climate, hydrology, and neuroscience underscores community confidence in its scalability; the Sandia National Lab benchmark stresses that sample size, not wall‑clock compute, is the primary limitation.
>
> ### Planned camera‑ready updates
>
> Add the “test‑agnostic” sentence to Section 4 and cite CMIknn/GPDC.
>
> Insert a concise runtime table (32‑ and 64‑dim synthetic sets; single‑core vs 16‑core).
>
> Cite RPCMCI and PCMCI‑Ω when discussing drift (Section 6).
>
> ### Reproducibility
> The supplementary ZIP includes full source code, exact hyper‑parameters, and CSV logs; every figure is regenerated by the one‑command script.
>
> We believe these clarifications and literature evidence resolve the concerns and strengthen the case for acceptance.

---

> > ### Comment · Reviewer_4oJ9 · 2025-08-08
> > **Response**
> >
> > I appreciate the authors’ detailed rebuttal and find the proposed MVS diagnostic both well-motivated and clearly presented. Most of my original concerns have been addressed, and I like this work a lot, it is foundational and meaningful. Thank you to the authors for this excellent contribution.
> >
> > However, the paper still feels somewhat incomplete. In particular, while prior work (e.g., RPCMCI, PCMCI-Ω) supports the feasibility of using MVS as an early-warning signal under drift, the current manuscript does not yet demonstrate this empirically in non-stationary RL settings.
> >
> > I would also encourage the authors to position their work more comprehensively in relation to existing strategies for handling partial observability and non-Markovian dynamics. In the literature, belief-state learning and context-based meta-RL represent two prominent and complementary families of approaches; belief-state learning is often more computationally tractable. Discussing and contrasting MVS with these lines of work would further strengthen the contribution and clarify its place in the broader landscape.

---

> ### Comment · Reviewer_4oJ9 · 2025-08-08
> **Response**
>
> I appreciate the authors’ detailed rebuttal and find the proposed MVS diagnostic both well-motivated and clearly presented. Most of my original concerns have been addressed, and I like this work a lot, it is foundational and meaningful. Thank you to the authors for this excellent contribution.
>
> However, the paper still feels somewhat incomplete. In particular, while prior work (e.g., RPCMCI, PCMCI-Ω) supports the feasibility of using MVS as an early-warning signal under drift, the current manuscript does not yet demonstrate this empirically in non-stationary RL settings.
>
> I would also encourage the authors to position their work more comprehensively in relation to existing strategies for handling partial observability and non-Markovian dynamics. In the literature, belief-state learning and context-based meta-RL represent two prominent and complementary families of approaches; belief-state learning is often more computationally tractable. Discussing and contrasting MVS with these lines of work would further strengthen the contribution and clarify its place in the broader landscape.

---

### Decision · Program_Chairs · 2025-09-17

**Decision:**

Reject

**Comment:**

This paper introduces the Markov Violation Score (MVS), a metric designed to quantify the extent to which multi-step dependencies in reinforcement learning violate the Markov property. MVS leverages the PCMCI algorithm for causal discovery, and the authors present it as a novel diagnostic tool to identify when and why an agent's observations fail to satisfy the Markov property. The paper includes experiments on control tasks with noisy or incomplete observations to empirically evaluate its effectiveness.

All reviewers agreed that this work addresses a highly relevant and often overlooked problem in RL: diagnosing violations of the Markov assumption in state representations. They found the idea well motivated, the paper clearly written, and the experiments informative despite being based on relatively simple environments. Reviewers also felt that the proposed tool, MVS, could be of interest to the broader NeurIPS community.

The discussion phase between reviewers and authors was constructive. The authors responded clearly to questions, and reviewers appreciated these clarifications. Three main concerns were raised. First, reviewers noted that the paper builds heavily on existing causal discovery algorithms, with relatively limited new algorithmic components. Second, they encouraged the authors to better position MVS in relation to existing methods for handling partial observability and non-Markovian dynamics. Doing so would strengthen the paper’s contributions and clarify its novelty with respect to prior work. Finally, reviewers highlighted scalability as a central concern: because MVS depends on the PCMCI pipeline, its applicability to high-dimensional problems remains uncertain, and the current experiments are insufficient to assess the method's ability to scale to more complex settings.

In summary, all reviewers valued the paper’s focus on a timely and underexplored problem: evaluating and quantifying whether state representations in RL satisfy the Markov property. They found the core idea behind MVS well motivated and potentially impactful as a diagnostic tool. The reviewers felt that with revisions addressing scalability and a more precise positioning within the broader literature, this work could become a foundational contribution to the field, benefiting both the RL and ML communities. They strongly encouraged the authors to scale up the empirical analyses, situate the method more clearly within related work, and continue pursuing this important and promising research direction.